# Mitogen-Activated Protein Kinases Associated Sites of Tobacco Repression of Shoot Growth Regulates Its Localization in Plant Cells

**DOI:** 10.3390/ijms23168941

**Published:** 2022-08-11

**Authors:** Luyao Wang, Ying Gui, Bingye Yang, Wenpan Dong, Peiling Xu, Fangjie Si, Wei Yang, Yuming Luo, Jianhua Guo, Dongdong Niu, Chunhao Jiang

**Affiliations:** 1Department of Plant Pathology, College of Plant Protection, Nanjing Agricultural University, Nanjing 210095, China; 2Key Laboratory of Integrated Management of Crop Disease and Pests, Ministry of Education, Nanjing 210095, China; 3Key Laboratory of Integrated Pest Management on Crops in East China, Ministry of Agriculture, Nanjing 210095, China; 4Key Laboratory of Plant Immunity, Nanjing Agricultural University, Nanjing 210095, China; 5Engineering Center of Bioresource Pesticide in Jiangsu Province, Nanjing 210095, China; 6Shenzhen Branch, Guangdong Laboratory of Lingnan Modern Agriculture, Agricultural Genomics Institute at Shenzhen, Chinese Academy of Agricultural Sciences, Shenzhen 518120, China; 7Shenzhen Branch, Genome Analysis Laboratory of the Ministry of Agriculture and Rural Affairs, Agricultural Genomics Institute at Shenzhen, Chinese Academy of Agricultural Sciences, Shenzhen 518120, China; 8Jiangsu Key Laboratory for Eco-Agricultural Biotechnology around Hongze Lake, Jiangsu Collaborative Innovation Center of Regional Modern Agriculture and Environmental Protection, Huaiyin Normal University, Huai’an 223300, China

**Keywords:** MAPK, phosphorylation, RSG, abiotic stress, protein interaction

## Abstract

Plant defense and growth rely on multiple transcriptional factors (TFs). Repression of shoot growth (RSG) is a TF belonging to a bZIP family in tobacco, known to be involved in plant gibberellin feedback regulation by inducing the expression of key genes. The tobacco calcium-dependent protein kinase CDPK1 was reported to interact with RSG and manipulate its intracellular localization by phosphorylating Ser-114 of RSG previously. Here, we identified tobacco mitogen-activated protein kinase 3 (NtMPK3) as an RSG-interacting protein kinase. Moreover, the mutation of the predicted MAPK-associated phosphorylation site of RSG (Thr-30, Ser-74, and Thr-135) significantly altered the intracellular localization of the NtMPK3-RSG interaction complex. Nuclear transport of RSG and its amino acid mutants (T30A and S74A) were observed after being treated with plant defense elicitor peptide flg22 within 5 min, and the two mutated RSG swiftly re-localized in tobacco cytoplasm within 30 min. In addition, triple-point mutation of RSG (T30A/S74A/T135A) mimics constant unphosphorylated status, and is predominantly localized in tobacco cytoplasm. RSG (T30A/S74A/T135A) showed no re-localization effect under the treatments of flg22, *B. cereus* AR156, or GA_3_, and over-expression of this mutant in tobacco resulted in lower expression levels of downstream gene *GA20ox1*. Our results suggest that MAPK-associated phosphorylation sites of RSG regulate its localization in tobacco, and that constant unphosphorylation of RSG in Thr-30, Ser-74, and Thr-135 keeps RSG predominantly localized in cytoplasm.

## 1. Introduction

Developmental plasticity is a unique feature of plants, and multiple complex programs evolved to adapt to different environmental situations. Plant hormones are known to contribute to innate transcriptional processes in specialized manners and to transfer exterior environmental stress to plant nuclei. Gibberllins (GAs) are tetracyclic diterpenoid phytohormones that regulate several aspects of plant growth and development, such as seed germination, stem elongation, leaf expansion, flowering, and fruiting [1]. Previous studies have reported repression of shoot growth (RSG) as a transcriptional activator in tobacco, which regulates endogenous amounts of GAs by controlling the expression of GA biosynthetic enzyme [2]. RSG was also reported to interact with the 14-3-3 protein, which works by binding to phosphorylated client proteins to modulate their function and form a highly conserved family of homodimeric and heterodimeric protein complex [3]. It has also been suggested that 14-3-3-signaling proteins could suppress RSG by restricting it in the cytoplasm of tobacco cells, and phosphorylation on Ser-114 of RSG was proven to be important for its binding with 14-3-3 [3]. In addition, the mutation of Ser-114 to Ala in RSG blocks its interaction with 14-3-3, and this mutated version of RSG (S114A) showed predominant localization in the nucleus after GA application [3]. A tobacco calcium-dependent protein kinase (CDPK1) was reported to decode calcium signal produced by GAs in tobacco and regulate intracellular localization of RSG. It is also suggested that CDPK1 was identified as an RSG kinase, which facilitates interaction between RSG and 14-3-3 [4].

MAPKs are involved in highly conserved signaling pathways in eukaryotes, and studies on plant MAPKs have attracted great attention, as MAPKs were key signal mediators in plants during response to different environmental stresses [5,6,7,8]. In previous reports, triggered by environmental signals, MAPK kinase kinase, named MAPKKK or MEKK, was firstly activated; then resulted in the phosphorylation of MAPK kinase, known as MAPKK, MEK, or MKK; and finally MAPK was activated. At the end of the phosphorylation-activating steps, MAPKs bind with and phosphorylate an amount of protein substrates to transfer upstream signals [8,9,10,11]. One well-studied MAPK, *Arabidopsis* MPK3, plays important roles in numerous processes, including defense against pathogens [5,10,12,13]. For example, the tobacco MPK3 (NtMPK3), also known as SIPK, is an ortholog of *Arabidopsis* MPK3, which is reported to be activated in response to fungal infection and increase production camalexin [14].

As is known, the closest protein homologue of RSG in *Arabidopsis thaliana* is VIP1, which is known to interact with *Agrobacterium* effector VirE2 and mediate nuclei transport of T-complex during *Agrobacterium*-mediated genetic transformation [15]. Interestingly, VIP1 was shown to interact with *Agrobacterium*-induced mitogen-activated protein kinase (MAPK) MPK3, and VIP1 relocalized from the cytoplasm to the nucleus upon phosphorylation by MPK3. Djamei et al. also reported that MAPK-dependent phosphorylation of VIP1 is necessary for *Agrobacterium*-mediated T-DNA transformation, and flg22, as a microbe-associated molecular pattern, is able to cause VIP1 to relocalize to the nucleus [16]. Moreover, a DNA hexamer motif named VIP1 response element (VRE) was identified. By binding to VREs, VIP1 regulates the expression of MPK3 pathway-related genes, such as *Trxh8* and *MYB44* [17]. Additionally, as a bZIP transcriptional factor in *A. thaliana,* VIP1 and its close homologues also showed liable localization in plant cells under abiotic stresses, such as hypo-osmotic stress, which mimics touch stimuli [18,19].

Considering the complexity of crosstalk throughout plant hormone pathways, it is meaningful to see whether RSG also interacts with other protein phosphokinases that are involved in different pathways, and whether artificial mutation on potential phosphorylation sites of RSG would interfere with its downstream functions as TF. In this study, we firstly identified a tobacco MPK3 (NtMPK3) as an RSG interaction protein kinase. By mutating potential MAPK-associated phosphorylation sites in RSG, NtMPK3-RSG interaction complex displayed different intracellular localization. Furthermore, the triple mutation in all three MAPK-associated phosphorylation sites (Thr-28, Ser-74, and Thr-135) to alanine completely blocked nuclear localization of RSG in tobacco cells. Moreover, we also report on the role of MAPK-associated phosphorylation sites as key regulators of RSG localization and its response to biotic or abiotic stresses. This work gives innovative insight into the molecular mechanism and whether MAPK pathway is linked to other plant hormone pathway regulators.

## 2. Results

### 2.1. Bacillus cereus AR156 Induces Nuclear Transport of RSG

As an ortholog of VIP1 in tobacco, RSG showed 48% exact (177/370) and 56% positive (210/370) hits with *Arabidopsis* VIP1 in protein sequence alignment, and both RSG and VIP1 had a conserved basic leucine zipper (bZIP) domain (Figure 1). By directly applying *B. cereus* AR156 [20,21,22] suspension on transgenic tobacco, which transiently expressed RSG with CFP tag, RSG started to re-localize in the tobacco nucleus within 5 min, and pre-dominant nuclear localization was observed within 30 min (Figure 2). These results indicate that, besides CDPK1, MAPK-mediated phosphorylation might also be potentially involved during intracellular relocalization of RSG under biotic stresses.

### 2.2. NtMPK3 Interacts with RSG as a Potential RSG Kinase

MAPKs, as terminal protein kinases in plant mitogen-activated protein kinase cascades, directly interact with targeted substrate proteins and continue with the phosphorylation process [11]. To verify the relationship between RSG and MAPKs in plants, we firstly investigated the interaction between RSG and NtMPK3 (XP_016478177.1), which was considered one of the representative tobacco MAPKs in this study. As shown in Figure 3A,B, we co-expressed RSG fused with nCerulean and NtMPK3 fused with cCFP in *N. benthamiana* leaves under the control of cauliflower mosaic virus 35S promoter (CaMV 35S), and the interaction between RSG and NtMPK3 was observed 3 days after *A. tumefaciens*-mediated infiltration. By over-expressing DsRed fused with the previously identified nuclear localization site (NLS) of VirD2 as a nuclear indicator [23], the RSG-NtMPK3 interaction complex showed a nuclear localizing pattern in *N. bent**hamiana* cells (20-times objective lens with 426–450 fluorescence filter) (Figure 3B). In addition, the interaction between RSG and MPK3 was conservative, and the interaction between RSG and AtMPK3 was observed as well. As the results show in Appendix A, RSG-AtMPK3 interaction complex also showed a nuclear localizing pattern in *N. bent**hamiana* cells (Appendix A).

Yeast-two-hybrid assay was then conducted to confirm the interaction between NtMPK3 and RSG in a yeast cell system. As NtMPK3 showed irrepressible self-activation in yeast-two-hybrid assay, the kinase interaction motif (KIM) docking site domain (91-342 aa) of NtMPK3 was identified and cloned into pSTT91. We expressed the KIM domain of NtMPK3 fused with LexA-BD and RSG fused with GAL4-AD in *Saccharomyces cerevisiae* strain TAT7 (L40-ura3), in which cell growth on a histidine-deficient medium indicates interaction. It was found that the KIM docking site domain of NtMPK3 interacted with RSG (Figure 3C), and the interaction ability of RSG was specific because it was not observed with the cell-to-cell movement protein (MP) of the *Tobacco mosaic virus* (TMV), which was considered an unrelated control protein. It is also revealed that the KIM docking site domain of NtMPK3 was the only specific domain to take charge of a binding protein substrate such as RSG (Appendix A).

BiFC and yeast-two-hybrid results were then validated by an independent approach, in which GFP-tagged NtMAPK3 was co-expressed in *N. benthamiana* leaves with c-Myc-tagged RSG, and isolated total protein was immunoprecipitated with anti-GFP antibody followed by Western blot analysis. Figure 4D shows that collected immunoprecipitates included both NtMPK3 detected by anti-GFP and RSG detected by anti-Myc, which confirms the interaction between RSG and NtMPK3 in the yeast-two-hybrid system and BiFC assay (Figure 4D).

Previous studies demonstrated that the AtVIP1 is a MAPK-associated protein substrate in *A. thaliana* [16]. To determine whether RSG, as an orthologue in tobacco, is a protein substrate of NtMPK3, we isolated protein extract from the *N. tobacum* plant over-expressing the RSG-CFP fusion protein, and Western blot analysis with monoclonal anti-GFP antibody detected additional shifted bands (Figure 3D). As the results show in Figure 3D, the λ-protein phosphatase (λ-PPase) treatment decreased the relative intensity of shifted bands, which indicates that these shifted bands correspond to the phosphorylated status of RSG-CFP fusion. Collectively, these results in Figure 3 support the notion that RSG not only interacts with NtMPK3 both in vitro and in vivo, but also is a protein substrate of NtMPK3 during the phosphorylation process.

### 2.3. MAPK-Associated Phosphorylation Sites in RSG Regulates Its Intracellular Localization with NtMPK3

A previous study identified Ser-114 in RSG, which was a phosphorylation site of an NtCDPK1, manipulating the interaction between 14-3-3 and RSG [4]. The results suggested that impaired phosphorylation on Ser-114 of RSG results in the reduction of expression of RSG-targeted genes [3,24]. To determine whether MAPK-associated phosphorylation sites of RSG were also involved in its downstream biological functions, we generated a set of potential mutant versions of RSG in which MAPK-associated phosphorylation sites were altered from Ser or Thr to Ala to simulate durable non-phosphorylation status, which were recorded as RSG^T28A^, RSG^T30A^, RSG^S74A^, and RSG^T135A^ (Figure 4). Interestingly, all of these generated RSG mutations showed interaction with the KIM docking site domain of NtMPK3 in yeast-two-hybrid assay, whereas only RSG^T28A^ still co-localized with NtMPK3 in the nucleus of *N. benthamiana* cells (Figure 4A,B).

Unlike their co-localization with NtMPK3, RSG mutants displayed both nuclear and cytoplasm localization in *N. benthamiana* cells by agro-infiltration (Figure 4C). However, nuclear localization of RSG^T30A^-CFP, RSG^S74A^-CFP, and RSG^T135A^-CFP was lackluster, although RSG^T28A^-CFP showed brighter nuclear localization (Figure 4C). Consequently, it was shown that Thr-30, Ser-74, and Thr-135 might play a more important role during NtMPK3-associated phosphorylation of RSG.

### 2.4. RSG^T30A^ and RSG^S74A^ Show Durable Localization in Tobacco Nuclear after B. cereus AR156 Treatment

It has been reported that the *Agrobacterium* treatment induced nuclear localization of AtVIP1 in *Arabidopsis* cells [16]. Thus, transgenic tobacco lines that stably express RSG^T30A^-CFP and RSG^S74A^-CFP driven by 35S promoter were generated to avoid possible biotic stimulation from *Agrobacterium* during agro-infiltration. Unlike the observed results in Figure 4C, stably expressed RSG^T30A^-CFP and RSG^S74A^-CFP displayed mere cytoplasm localization in *N. tobacum* cells, in which *Agrobacterium* stimulation was excluded (Figure 5).

To investigate whether NtMPK3 activation could affect the intracellular localization of RSG in tobacco cells, flg22, a well-studied peptide corresponding to conserved flagellin sequence of pathogenic bacteria that initiate plant MAPK-signaling cascades [16], and *B. cereus* AR156 cell suspension, were applied on acquired transgenic tobacco leaves (Figure 5A and 5B). Upon treatment of flg22, we observed strong increased signals of RSG-CFP, RSG^T30A^-CFP, and RSG^S74A^-CFP in the tobacco nucleus in 5 min. However, RSG^T30A^-CFP and RSG^S74A^-CFP re-localized in the cytoplasm in 30 min, whereas RSG-CFP retained significant nuclear localization (Figure 5A). Interestingly, *B. cereus* AR156, which was demonstrated to be an effective plant defense elicitor previously, facilitated lasting nuclear transport of RSG-CFP, RSG^T30A^-CFP, and RSG^S74A^-CFP in 30 min, which might have been due to *B. cereus* AR156 providing more intensive and continuous stimulation to plants than flg22 (Figure 5B). Taken together, these results support the notion that MAPK-associated phosphorylation sites of RSG are also involved in its intracellular re-localization. 

### 2.5. Mimic Constant MAPK-Associated Unphosphorylation in RSG Leads Pre-Dominant Cytoplasm Localization

As shown in previous research in VIP1, Ser-79, a MPK3-targeted phosphorylation site, was mutated to Asp to obtain constant phosphorylated VIP1, and this VIP1 mutation showed pre-dominantly [16]. In contrast to preventing phosphorylation on MAPK-associated sites in RSG, Thr-30, Ser-74, and Thr-135 in RSG were mutated to alanine to mimic constitutively non-phosphorylated status, which was marked as RSG^3M^. Interestingly, RSG^3M^ shows predominant cytoplasm localization in *N. benthamiana* by agro-infiltration (Figure 6A). To better evaluate the difference between RSG^3M^ and other RSG mutants and exclude the stimulation from *Agrobacterium*, we generated transgenic *N. tobacum* overexpressed RSG^3M^ fused with CFP tag. Ishia et al. reported that RSG transports to the plant nucleus in response to the reduction of plant GA contents, and exogenous GA treatment could lead the RSG cytoplasm re-localization [3]. A related study also revealed that CDPK1-dependent pathway is involved in GA-induced RSG nuclear export, and phosphorylation of Ser-114 was specifically important in this process [4]. In our study, apparent nuclear export was also observed 3 h after GA3 treatment in transgenic tobacco, which over-expresses RSG-CFP (Figure 6C). RSG mutants, including alanine substitution of Thr-30 or Ser-74 and alanine substitution of Thr-30, Ser-74, and Thr-135, showed stable predominant cytoplasm localization both in pre- and post-GA treatments (Figure 6C). As shown in Figure 6B, RSG^3M^ naturally localized in tobacco cytoplasm, and no significant nuclear relocalization was observed, although flg22 and *B. cereus* AR156 were applied (Figure 6B).

Collectively, this part of the results identified a triple amino acid mutant of RSG that maintains stable cytoplasm intracellular localization, and MAPK-associated phosphorylation sites in RSG might also participate in GA-regulated RSG nuclear export in tobacco cells.

### 2.6. Constant MAPK-Associated Phosphorylation in RSG Reduces Expression of Downstream Genes

RSG is also a functional transcriptional factor in tobacco, and a GA biosynthetic pathway related gene *NtGA20ox1* was reported to be a direct target of RSG [2]. To examine whether MAPK-associated phosphorylation sites of RSG regulate its function as transcription factor in GA biosynthetic pathway, we analyzed the expression level of *NtGA20ox1* in transgenic tobacco plants that overexpress RSG or RSG mutants (Figure 7A,B). By PCR and qPCR assays, the highest expression level of *NtGA20ox1* was detected in RSG over-expressed tobacco, and the lowest expression of *NtGA20ox1* was detected in RSG^T30A^, RSG^S74A^, and RSG^3M^ over-expressed tobacco (Figure 7A,B). The expression fold change of *NtGA20ox1* in transgenic tobacco over-expressing RSG, RSG^T30A^, RSG^S74A^, and RSG^3M^ were 1.56, 0.37, 0.55, and 0.50 compared with non-transgenic lines, respectively. These results, shown in Figure 7A,B, are consistent with the data presented in Figure 5A and Figure 6A, in which alternations of MAPK-associated phosphorylation sites in RSG significantly led to its pre-dominant cytoplasm localization.

We further compared the GA contents in acquired transgenic tobacco over-expressing RSG or its mutants with wild-type tobacco (Figure 7C). Interestingly, RSG over-expressing tobacco maintained the lowest GA contents compared with wild-type tobacco and other transgenic tobacco plants (Figure 7C). This might be due to large amount of RSG accumulation in the tobacco nuclei, which further triggered negative feedback during GA biosynthesis. These results suggest that MAPK-associated phosphorylation sites in RSG not only regulate its intracellular localization, but also manipulate its biological functions, and over-expressing its related variants would further alter the germination rate of tobacco seeds. Interestingly, the germination rate of transgenic tobacco seeds was as low as 43.1% (44 seeds germinated out of a total 102 seeds) and recovered to 81.7% (89 seeds germinated out of a total 107 seeds) and 75.4% (80 seeds germinated out of a total 106 seeds) when Thr-30 or Ser-74 was mutated to alanine. The transgenic tobacco seeds over-expressing RSG^3M^ only produced weak and growth-aborting seedlings at 18.5% (19 seeds germinated out of a total 103 seeds) of the germination rate (Figure 7D).

## 3. Discussion

### 3.1. Biological Functions of RSG in Tobacco and VIP1 in Arabidopsis

Tobacco transcription factor RSG was first identified in tobacco with the function of regulating morphology of plants by controlling the endogenous amounts of GAs [25]. It was then discovered to control the feedback regulation of NtGA20ox1 via intracellular localization and epigenetic mechanism, and signaling protein 14-3-3 was proven to be an RSG binding partner and participate in nuclear–cytoplasmic shuttling [24,26]. Tobacco calcium (Ca^2+^)-dependent protein kinase CDPK1 (NtCDPK1) was identified as an RSG kinase that promotes 14-3-3 binding of RSG by phosphorylation of RSG [4,27]. Our proposed involvement of RSG interacting with MPK3 is consistent with VIP1, an *Arabidopsis* homologue of RSG, in previous research [16]. Interestingly, VIP1 showed several similarities with RSG: VIP1 presented a nuclear–cytoplasmic shuttling subcellular localization pattern [28], and VIP1 interacted with 14-3-3 and its subcellular localization was regulated by phosphorylation [29]. *B. cereus* AR156 is an effective biocontrol agent, which was previously isolated from the natural environment [20]. *B. cereus* AR156 facilitates plant defense against a variety of diseases through SA- and JA/ET-dependent pathways, which is known as induced systematic resistance (ISR) [21]. It was also reported that *B. cereus* AR156 treatment could induce the MPK3/MPK6 pathway in *Arabidopsis* by biosynthesizing extracellular polysaccharides (EPS) [22]. The initial aim of this study was to investigate whether biocontrol agents, such as *B. cereus* AR156, also manipulate nuclear localization of functional transcriptional factors in either plant defense or the growth process. In previous research presented by Armin et al., *Arabidopsis* VIP1 relocalized from the cytoplasm to the nucleus and regulated the expression of defense-related genes, such as *PR1*, in the presence of flg22 or the *Agrobacterium* strain [16]. It was further revealed that VIP1 binds with specific DNA segments and promotes the expression of *Trxh8 and MYB44* to induce MPK3 pathway [17].

### 3.2. Intracellular Movement of RSG under Biotic Stresses

*Agrobacterium* is known to prime plant defense response [30]; thus, most of the experiments in our study were carried out on transgenic plants. In Figure 4C, subcellular localizations of the RSG mutant were observed by *Agrobacterium* infiltration on *N. benthamiana* leaves, and both cytoplasm and nuclear localization were observed for RSG^T28A^, RSG^T30A^, RSG^S74A^, and RSG^T135A^. In transgenic tobacco, RSG^T30A^ and RSG^S74A^ showed only cytoplasm localization before flg22 and *B. cereus* AR156 treatment, indicating that the *Agobacterium* infiltration process actually stimulates nuclear transport of the RSG protein and related mutants (Figure 5). HXRXXS motifs were demonstrated as 14-3-3 binding targets [31], and S114 in RSG, S35, S115, and S151 in VIP1 were identified as HXRXXS motifs. The mutated RSG on the S114 site to alanine could not bind to 14-3-3 and exclusively localized in the nucleus, and the non-phosphorylated VIP1 mutant of three HXRXXS motifs showed no interaction with 14-3-3 and only localized in the nucleus [24,29]. Importantly, the four predicted MAPK-mediated phosphorylation sites of RSG did not correspond to the pattern of the HXRXXS motif (Figure 1), indicating that a blocking interaction with 14-3-3 might not be the cause of the changes in nuclear–cytoplasmic localization of mutated RSG in this study. Moreover, artificial mutating MAPK-associated phosphorylation sites of RSG to alanine resulted in only cytoplasmic protein localization, which was contrary to the dominant nuclear localization of RSG mutated on the NtCDPK1-associated site [24]. One possible explanation is that RSG, as an important bZIP transcription factor, might be phosphorylated by different protein kinases on different motifs under different stimulations, for example, abiotic and biotic stresses. On the other hand, in our results, biotic stimulation, for example, bacterial flagellin peptide flg22, induced nuclear import of RSG up to 30 min (Figure 5A). However, on transgenic tobacco overexpressing RSG^T30A^ and RSG^S74A^, mutated RSG rapidly re-localized back to the cytoplasm 30 min post flg22 treatment, although nuclear import was observed in the initial 5 min. Single mutation of MAPK-associated sites partly impaired the nuclear import of RSG under biotic stimulation, as flg22 was proven to trigger activation of MAPK cascades in plants [17]. Biocontrol agent strain *B. cereus* AR156 is also reported to prime plant defense, and MAPK-signaling pathway is involved in this process as well [21,32]. Interestingly, continuous nuclear transport of RSG^T30A^ and RSG^S74A^ was detected on transgenic tobacco leaves treated with *B. cereus* AR156 in 30 min (Figure 5B). The reason might be that, unlike flg22, *B. cereus* AR156 cells might provide efficient and constant biotic stimulation on treated plant tissues. Although one MAPK-associated phosphorylation site was mutated, phosphorylation on the rest sites (Ser-74, Thr-135 in RSG^T30A^ and Thr-30, Thr-135 in RSG^S74A^) would still contribute to nuclear transport of RSG under biotic stresses, and stronger stimulation might trigger a longer retention time of RSG in plant nuclei. Indeed, *B. cereus* AR156 was proven to prime plant defense in several ways, including secreting extracellular polysaccharides and manipulating specific miRNA or transcription factors in treated plants [22,32,33].

### 3.3. Biological Roles of MAPK-Associated Phosphorylation Sites in RSG

To better illustrate the importance of MAPK-associated phosphorylation sites in RSG during nuclear–cytoplasmic shuttling under biotic stresses, we generated a triple-point mutant of RSG on three MAPK-associated phosphorylation sites to alanine, including Thr-30, Ser-74, and Thr-135, and it resulted in RSG^3M^. Consistent with single-mutant RSG^T30A^ and RSG^S74A^, RSG^3M^ showed only cytoplasm localization in transgenic plants before treatment, whereas neither flg22 nor *B. cereus* AR156 treatment could induce nuclear transport of RSG^3M^ (Figure 7A,B). These data further manifest the importance of MAPK-associated phosphorylation sites of RSG in response to biotic stresses in plants. The nuclear localization of RSG is directly related to its function as a transcription factor. The downstream gene-regulating capacity of RSG plays an important role in GA feedback regulation in tobacco, and *NtGA20ox1* was reported to be a downstream gene regulated by RSG [4,26]. Reduction of *NtGA20ox1* expression level in transgenic tobacco of RSG^T30A^, RSG^S74A^, and RSG^3M^ was consistent with predominant cytoplasm localization of these RSG mutants. As a previous study demonstrated RSG could interact with itself like other plant bZIP family proteins, over-expression of a point-mutated version of RSG might hijack endogenous RSG and block its native functions [25]. However, we did not detect significantly higher GA content in transgenic tobacco of RSG^T30A^, RSG^S74A^, or RSG^3M^ compared with wild-type plants, which might be because of the complicated plant hormone feedback-regulating mechanisms. Specifically, germination of seeds obtained from RSG^3M^ transgenic tobacco was relatively slow and incomplete compared with seeds obtained from RSG, RSG^T30A^, and RSG^S74A^ over-expression in tobacco, suggesting that the function of RSG as a transcription factor is much more important for the tobacco seed-germinating stage. 

Collectively, the interaction between RSG and NtMPK3, the involvement of MAPK-associated phosphorylation sites in nuclear–cytoplasmic shuttling of RSG, and observed phenotype variations in transgenic plants over-expressing the mutated version of RSG suggest that phosphorylation on MAPK-associated sites on RSG deeply regulate its function in plants. Interestingly, RSG was previously proven to bind VIP1 response element (VRE) sequence and activate downstream gene expression [34], indicating potential target gene-range overlaps between RSG and VIP1. It would be particularly interesting to investigate further functions of RSG in downstream defense response against plant pathogens. 

## 4. Material and Methods

### 4.1. Bacterial Strains, Plants, and Growth Conditions

*Agrobacterium tumefaciens* strain EHA105 and *Escherichia coli* strain DH5α were grown in Luria–Bertani (LB) agar (NaCl 10 g·L^−1^, yeast extract 5 g·L^−1^, tryptone 5 g·L^−1^) at 28 °C overnight. *Bacillus cereus* AR156 was isolated from the forest soil of Zhenjiang City, Jiangsu Province, China, as an effective bacterial BCA (Genebank accession number CP015589) and grown overnight in liquid LB broth at 30 °C in a shaking incubator. *Nicotiana tabacum* var. Turk and *N. bent**hami**ana* plants were grown in soil or in MS medium (MES 0.5 g·L^−1^, sucrose 30 g·L^−1^, agar 8 g·L^−1^, pH 5.8) after seed-surface sterilization and maintained in vitro. All plants were grown in environment-controlled growth chambers under long-day conditions (16 h light/8 h dark cycle at 140 µE sec^−1^m^−2^ light intensity) at 22 °C.

### 4.2. Plasmid Construction and Mutagenesis

Plasmids and cloning strategies are summarized in Appendix A, and the primer sequences used in these cloning procedures are described in Appendix A. For instance, for Gal4-AD fusions, the coding sequences of RSG and its variants were PCR amplified by primer pair 1F/1R, using total *N. tabacum* cDNA library as template, and cloned into the indicated sites of pGAD424 (LEU^2+^, Clontech; Mountain View, CA, USA). For LexA fusions, the coding sequences of NtMPK3 and AtMPK3 were PCR-amplified with indicated primer pairs 2F/2R and 3F/3R, using the total *N. tabacum* cDNA library as a template, and cloned into the indicated sites of pSTT91 (TRP^1+^) [35]. For generating point mutations in RSG, overlapping PCR reactions were conducted to generate codon substitutions. Two DNA segments were amplified by PCR: from the translation initiation codon at the 5′-end to the target codon position and from the target codon position to the 3′ end of the coding sequence. For example, to generate the T28A mutation in RSG, PCR reactions were performed as previously described [34], with the primer pairs 1F-11R and 11F-1R (Appendix A). The two PCR products then were used as a template to generate the full-length mutated RSG sequence by overlapping PCR with the primer pair, followed by insertion of the resulting PCR product, which encodes the full-length RSG T28A mutant, into the *BamHI*/*PstI* site of plasmid pGAD424, resulting in pGAD424-RSG-T28A (Appendix A), and other RSG point mutations were introduced with primer pairs (12F/12R for T30A, 13F/13R for S74A, 14F/14R for T135A). Aiming to express targeted proteins in plants, the coding sequences of RSG were inserted into the multiple cloning sites of pSAT5-CFP-C1 for subcellular localization experiments or pSAT1-ncerulean-C1/pSAT5-cCFP-C1 for BiFC assays [36]. The resulting pSAT series expression cassettes were excised with AscI for pSAT1 or I-CeuI for pSAT5 and transferred to the same site of the binary pPZP-RCS2 or the pPZP-DsRedNLS-RCS vector with the DsRed signal fused with an NLS tag of VirD2 [37].

### 4.3. Yeast-Two-Hybrid Protein Interaction Assay

For yeast-two-hybrid experiments, pSTT91 and pGAD424 plasmids cloned with potential interactors were introduced into the *Saccharomyces cerevisiae* strain L40 and grown for 2 days at 30 °C on a leucine- and tryptophan-deficient synthetic defined premixed yeast growth medium (SD-Leu-Trp, TaKaRa Clontech, Kusatsu, Japan) [38]. Five to 10 acquired yeast colonies on SD-Leu-Trp plates were resuspended in sterilized water and plated on SD-Leu-Trp and the same medium plate lacking leucine, tryptophan, and histidine (SD-Leu-Trp-His). Yeast cell growth was observed after incubation at 30 °C for 2–3 days as previously described [34].

### 4.4. Biomolecular Fluorescence Complementation Assay

For biomolecular fluorescence complementation (BiFC) assay, tested protein pairs were fused with either nCerulean or cCFP tag and cloned into pSAT series plasmid [36]. For example, to observe the interaction between RSG and NtMPK3, the coding sequence of RSG was fused with nCerulean in pSAT1-nCerulean-C1, and the coding sequence of NtMPK3 was fused with cCFP in pSAT5-cCFP-C1. The resulting expression cassette of RSG was then inserted into the pPZP-RCS1 binary vector and the expression cassette of NtMPK3 was inserted into the pPZP-RCS1-DsRedNLS binary vector. These two constructs were transiently co-expressed in *N. benthamiana* by agroinfiltration. CFP and DsRed fluorescence signals were detected after 2 days with a CLSM (Leica AF6000 modular microsystems, Wetzlar, Germany) as previously described [34]. All experiments were biologically repeated three times.

### 4.5. Semi-Quantitative PCR

For semi-quantitative PCR assays, total RNA from the shoots of transgenic tobacco expressing RSG-CFP or its related mutations under the control of CaMV 35S promoter or control wild-type *N. tabacum* were converted into cDNA with a PrimeScript^TM^ RT reagent Kit (TaKaRa). PCR was performed with cDNA derived from 0.5 μg of total RNA with Extaq (TaKaRa). The primer sequences were 5′-CAACGCCCATCGTTTCATGG-3′ and 5′-CAAAAACTTGAAGCCCGCCA-3′ for NtGAox20 and 5′-ATGTGTTCGTTTCAGCCCGA-3′ and 5′-CCGCTGAAAAGTGTGCTTCC-3′ for tobacco *arcA* as an internal gene control as previously described [3].

### 4.6. Gibberellin Content Assay

The contents of endogenous gibberellin (GA) were detected with a Plant GA ELISA Kit (FEIYA Biotechnology, Guangzhou, China) with the double antibody sandwich method. Specifically, microwell plates were first coated with purified GA antibody, and plant tissue extraction and GA-HRP antibody were then added to the wells for 1 h antibody staining. After drastically washing, TMB (3,3′5, 5′-tetramethylbenzidine) was added to the wells as a reaction substrate. The color of the reaction solution was transformed to yellow in the presence of HRP at a low pH value, and the OD reads at 450 nm represented GA contents by referring to the standard curve.

### 4.7. Confocal Fluorescence Microscopy

CLSM (Leica AF6000 Modular Microsystems) was used to collect sample images. A 434 nm line from an argon ion laser was used to excite cyan fluorescent protein (CFP), and a 558 nm line from an argon ion laser was used to excite red fluorescent protein (DsRed) as previously described [34]. For each assay, six independent leaves were observed for each experiment and each experiment had at least repetitions. 

### 4.8. Statistical Analysis

One-way analysis of variance (ANOVA) was carried out with IBM SPSS (version 19.0, Armonk, NY, USA) and followed by Duncan’s multiple range tests (*p* < 0.05) for statistical analysis and Student’s *t*-test for evaluating the significance in all data.

## 5. Conclusions

In our research, we found that tobacco RSG was involved in gibberellin feedback regulation by inducing the expression of key genes, and we identified tobacco mitogen-activated protein kinase 3 (NtMPK3) as an RSG-interacting protein kinase. The mutation of the predicted MAPK-associated phosphorylation site of RSG significantly altered the intracellular localization of the NtMPK3-RSG interaction complex. Furthermore, in our study, it was also suggested that excessive accumulation in the nuclei or dominate accumulation in the cytoplasm of RSG would impair tobacco seed germination. Our results highlight the associations between MAPK-associated phosphorylation sites and biological functions of plant transcription factor RSG, one kind of bZIP-family protein in tobacco.

## Figures and Tables

**Figure 1 ijms-23-08941-f001:**
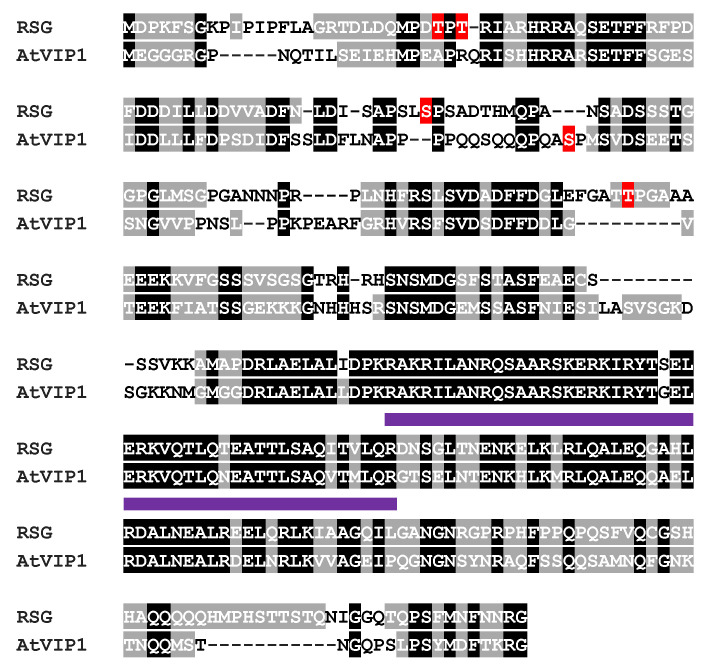
Amino acid sequence alignment of RSG and AtVIP1. The RSG from *N. tabacum* cv. *Turk* (accession number BAA97100.1) was aligned using the T-Coffee program (http://www.ebi.ac.uk/Tools/t-coffee/index.html, accessed on 25 June 2022) with its closest homolog AtVIP1 from *Arabidopsis thaliana* (AtVIP1, accession number NP_564486.1). Amino acid residues identical between RSG and AtVIP1 are highlighted in black and gray for conserved hits. The basic leucine zipper (bZIP) domains are underlined in purple. Amino acids potentially phosphorylated by MAPK are highlighted in red.

**Figure 2 ijms-23-08941-f002:**
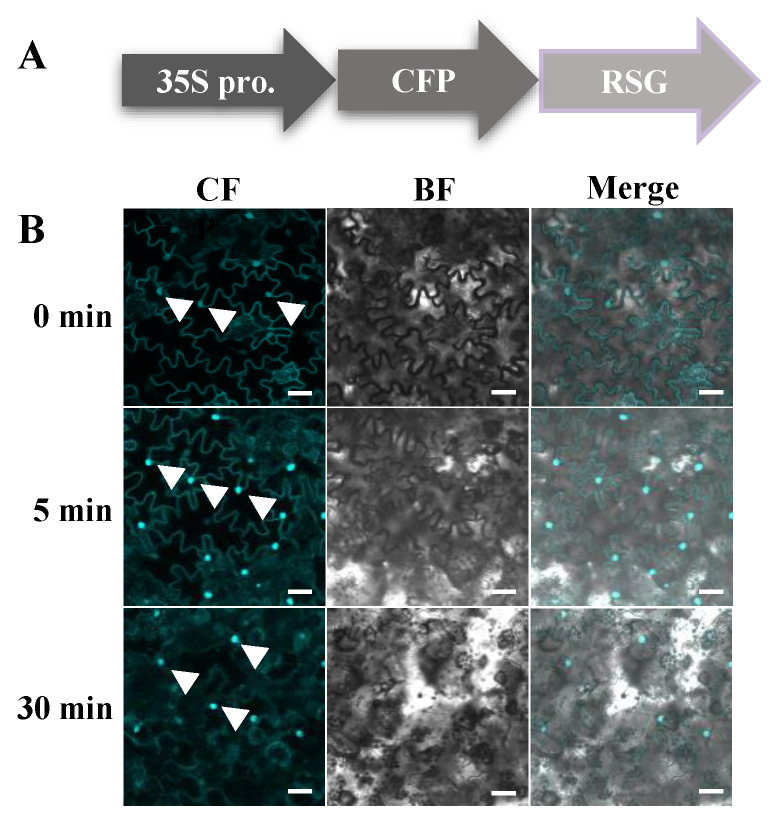
The *B. cereus* AR156 induces nuclear import of RSG in *N. tabacum* cells. (**A**) Constructs of 35S-driven RSG-CFP; (**B**) confocal microscopy observation of transgenic tobacco overexpresses RSG-CFP upon treatment of *B. cereus* AR156 suspension in 5 min and 30 min. RSG tagged with CFP were stably expressed in transgenic *N. tabacum* and analyzed by confocal microscopy after *B. cereus* AR156 treatment. CFP signal is in cyan. Images are single confocal sections, representative of images obtained in three independent experiments. Scale bars = 40 µM. White arrows indicate observed nuclear of tobacco cells. Scale bars = 20 µm. Three independent experiments were performed for each assay with similar results.

**Figure 3 ijms-23-08941-f003:**
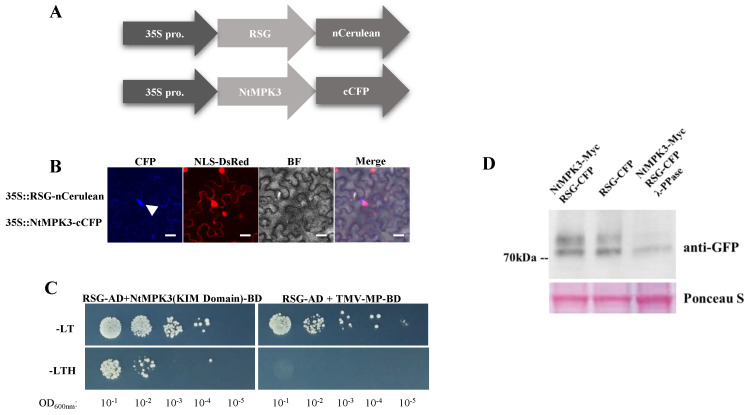
RSG interacts with mitogen-activated protein kinase 3 (MAPK3) in tobacco. (**A**) Constructs of 35S-driven RSG-nCerulean and 35S-driven NtMPK3-cCFP. (**B**) BiFC assay. RSG-nCerulean and NtMPK3-cCFP were transiently expressed in agroinfiltrated leaf epidermis of *N. benthamiana*. NLS-VirD2 fused with DsRed indicated the localization of the nucleus and was analyzed by confocal microscopy three days post-infiltration. CFP signal is in cyan. Images are single confocal sections, representative of images obtained in two independent experiments performed for each protein; for each experiment, three infiltrations were performed on three different leaves, with two images recorded per infiltration. Scale bars = 20 µM. (**C**) Yeast-two-hybrid interaction assay. LexA-NtMPK3(KIM Domain) was co-expressed with Gal4-AD fused to RSG. The indicated dilutions of cell cultures were plated and grown on non-selective (+histidine) and selective media (−histidine). (**D**) Detection of phosphorylation of RSG co-expresses with NtMPK3. For λ-PPase treatment, 10 units μL-1 λ-PPase were added to the reaction and incubated at 30 °C for 30 min. Three independent experiments were performed for each assay with similar results.

**Figure 4 ijms-23-08941-f004:**
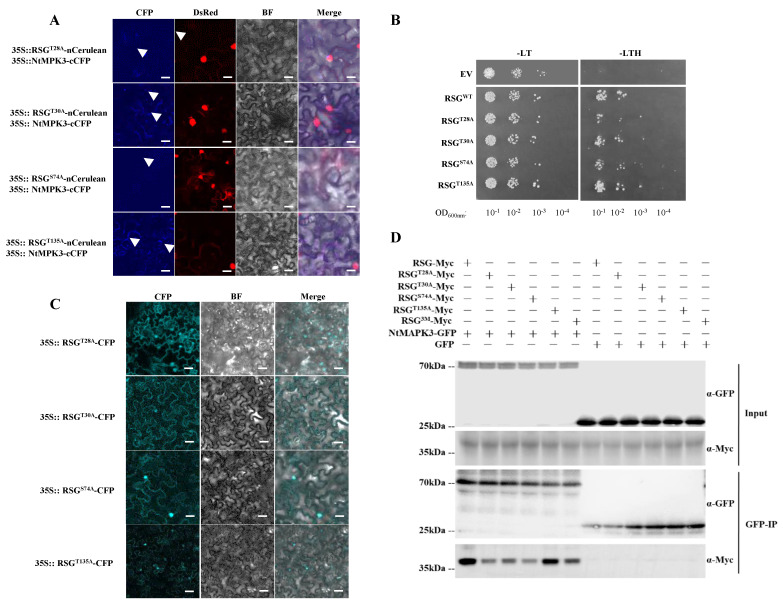
Interaction between NtMPK3 and four amino acid mutants of NtRSG and their localization in tobacco cells by BiFC. (**A**) BiFC assay. RSG-nCerulean or RSG-derived single-point amino acid mutants fused with nCerulean and NtMPK3-cCFP were co-expressed in *N. benthamiana*. NLS-VirD2 fused with DsRed refer to the localization of nuclear. Scale bars = 20 µm. (**B**) Yeast-two-hybrid interaction assay. LexA-NtMPK3(KIM Domain) was co-expressed with Gal4-AD fused to RSG or RSG site mutants. The indicated dilutions of cell cultures were plated and grown on non-selective (+histidine) and selective media (−histidine). (**C**) RSG and indicated mutants tagged with CFP were transiently expressed in agroinfiltrated leaf epidermis of *N. benthamiana* and analyzed by confocal microscopy three days post-infiltration. CFP signal is in cyan. Images are single confocal sections, representative of images obtained in two independent experiments performed for each protein; for each experiment, three infiltrations were performed on three different leaves, with two images recorded per infiltration. Scale bars = 40 µM. (**D**) Co-immunoprecipitation interaction assay. NtMAPK3-GFP was expressed with C-terminus Myc-tagged RSG and its relative mutants for 3 days in agroinfiltrated Nicotiana benthamiana leaves and immunoprecipitated (IP) with anti-GFP antibody (top panel), followed by Western blot analysis with anti-GFP or anti-Myc antibody. Two independent experiments were performed for each assay with similar results.

**Figure 5 ijms-23-08941-f005:**
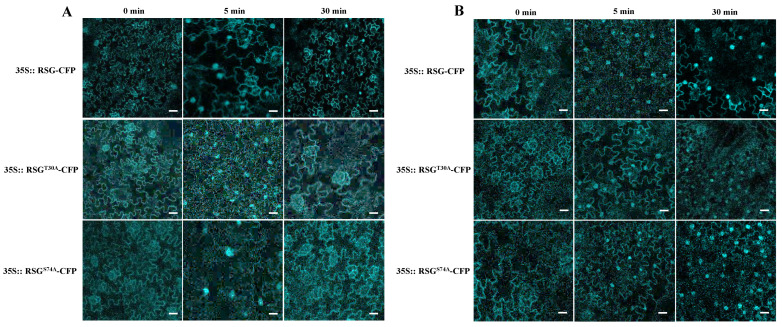
RSG^T30A^ and RSG^S74A^ show durable localization in tobacco nuclear after *B. cereus* AR156 treatment. (**A**) Observation of transgenic tobacco overexpressed RSG, RSG^T30A^, or RSG^S74A^ fused with CFP upon treatment of flg22. RSG and indicated mutants tagged with CFP were stably expressed in transgenic *N. tabacum* and analyzed by confocal microscopy after flg22. (**B**) Observation of transgenic tobacco overexpressed RSG, RSG^T30A^, or RSG^S74A^ fused with CFP upon treatment of *B. cereus* AR156 suspension. RSG and indicated mutants tagged with CFP were stably expressed in transgenic *N. tabacum* and analyzed by confocal microscopy after *B. cereus* AR156 treatment. CFP signal is in cyan. Images are single confocal sections, representative of images obtained in two independent experiments performed for each protein; for each experiment, three infiltrations were performed on three different leaves, with two images recorded per infiltration. Scale bars = 40 µM.

**Figure 6 ijms-23-08941-f006:**
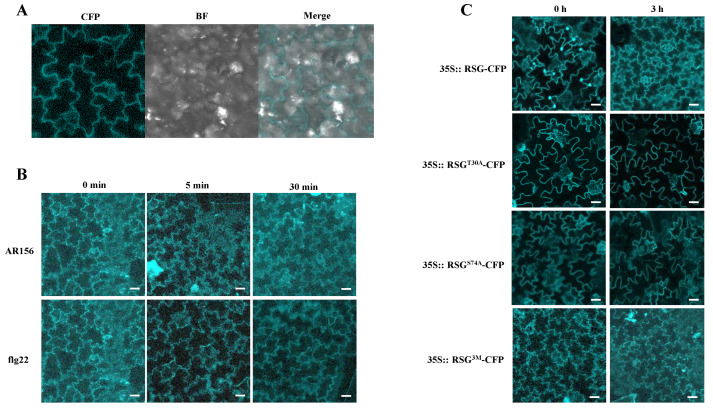
Mimicking non-MAPK-associated phosphorylation in RSG shows stable pre-dominant cytoplasm localization. (**A**) RSG 3M (T30D/S74D/T135D) tagged with CFP were transiently expressed in agroinfiltrated leaf epidermis of *N. benthamiana* and analyzed by confocal microscopy three days post-infiltration. (**B**) RSG^3M^ (T30A/S74A/T135A) tagged with CFP were stably expressed in transgenic *N. tabacum* and analyzed by confocal microscopy upon treatment of flg22 or *B. cereus* AR156 suspension. (**C**) Observation of transgenic tobacco overexpressed RSG, RSG^T30A^, RSG^S74A^, or RSG^3M^ fused with CFP upon treatment GA_3_. RSG and indicated mutants tagged with CFP were stably expressed in transgenic *N. tabacum* and analyzed by confocal microscopy 3 h after GA_3_ treatment. CFP signal is in cyan. Images are single confocal sections, representative of images obtained in two independent experiments performed for each protein; for each experiment, three infiltrations were performed on three different leaves, with two images recorded per infiltration. Scale bars = 40 µM.

**Figure 7 ijms-23-08941-f007:**
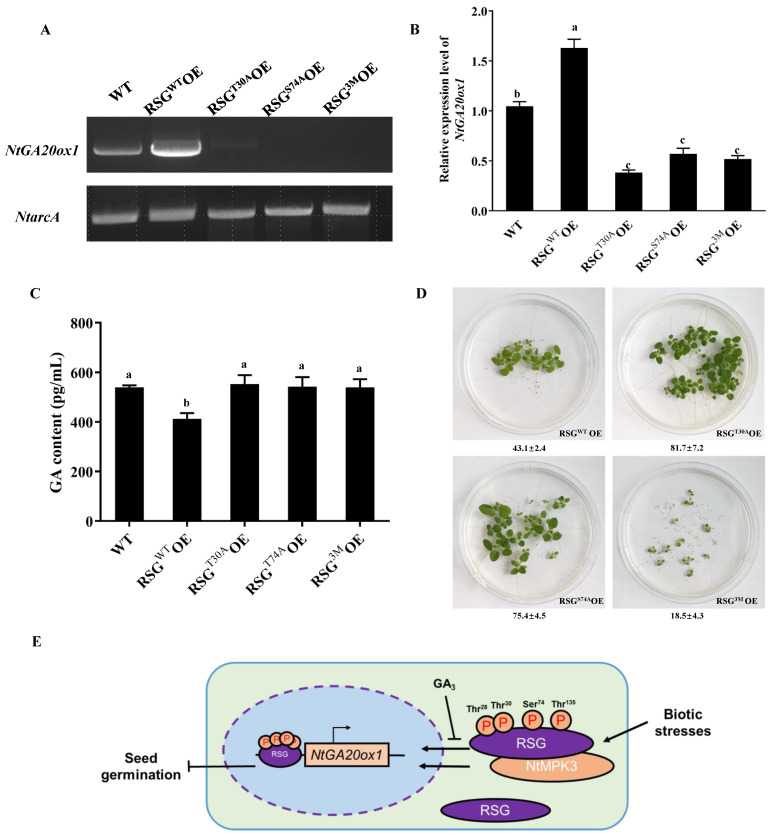
MAPK-associated phosphorylation sites in RSG regulates its function as a transcriptional factor. (**A**,**B**) Semi-quantitative PCR analysis of *NtGA20ox1* expression in transgenic tobacco. (**C**) Detection of GA contents in transgenic tobacco overexpresses RSG or its related mutants. The values are mean ± standard deviation, followed by the same letter within a column not being significantly different, as determined by Duncan’s multiple range test (*p* < 0.05). (**D**) Seeds of transgenic tobacco overexpressing RSG or its relative mutants were germinated in MS medium. The values at the bottom of panels indicate the germination ratio calculated from three independent plates. Three biological replicates were performed for each assay with similar results. (**E**) Schematic representation of the main idea of this research.

## Data Availability

All data generated or analyzed during this study are included in this published article.

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
