# Peer review of "Mitogen-Activated Protein Kinases Associated Sites of Tobacco Repression of Shoot Growth Regulates Its Localization in Plant Cells"

_ijms, 2022, doi:10.3390/ijms23168941_

Round 1
Reviewer 1 Report
The research focuses on RSG (Regression of Shoot Growth), a leucine zipper transcription factor involved in plant gibberellin feedback regulation. NtMPK3 (CDPK1) phosphorylates RSG at residues Thr-30, Ser-74, and Thr-135, some of which are necessary for RSG nuclear transport. Notably, nuclear transport of RSG, T30A, and S74A occurred with treatment by the flg22 plant defense elicitor, but the mutant versions returned to the cytoplasm within 30 min. These results infer that NtMPK3 is necessary for nuclear localization of RSG by flg22 for plant defensive function.
The experimentation is comprehensive and highly technical, and the results thoroughly and adequately support the stated conclusions in the manuscript. Criticism is that the research is not very hypothesis-based, and the investigation is an enormous biochemical and genetic tactical endeavor to decipher RSG function with associated partners molecules. Principle hypotheses about RSG’s critical biological functions in plant defense derive mainly from experimentation that utilizes heterologous biological systems for functional tests but can not directly establish cause and effect biology between the plant and the microbe. The yeast genetic experiments seem adequately conducted but not directly interpretable. The biomolecular fluorescence datasets do not confirm whether the sample differences are biologically meaningful as there are no statistically analyzed data or compelling text interpretation.
The writing is clear, making the research, results, and conclusions comprehensible. Albeit, the paper should be more hypothesis-based and substantial discoveries more evident.
Author Response
We appreciate the reviewer’s enthusiasm on our work. Thanks for the detailed comments for clarification and improvement of our manuscript. We have prepared a revised manuscript to follow the reviewer’s concerns. We have addressed all comments below.
1. The experimentation is comprehensive and highly technical, and the results thoroughly and adequately support the stated conclusions in the manuscript. Criticism is that the research is not very hypothesis-based, and the investigation is an enormous biochemical and genetic tactical endeavor to decipher RSG function with associated partners molecules. Principle hypotheses about RSG’s critical biological functions in plant defense derive mainly from experimentation that utilizes heterologous biological systems for functional tests but can not directly establish cause and effect biology between the plant and the microbe.
Response: First of all, thank you for your affirmation of this study. In this study, we identified tobacco mitogen-activated protein kinase 3 (NtMPK3) as a RSG interacting protein kinase. And demonstrated that MAPK associated phosphorylation sites of RSG regulate its localization in tobacco and constant unphosphorylation of RSG in Thr-30, Ser-74 and Thr-135 keeps RSG predominantly localized in cytoplasm. Indeed, the above findings and conclusions are mainly proved by biochemical research methods, lack of further biological verification, and we simplely discuss the impact of RSG by some immunity elicitor such as flg22, PGPR strain et al. Laterly, we will integrate more biological methods to study the function of related genes in the interaction between plants and pathogens.
2. The biomolecular fluorescence datasets do not confirm whether the sample differences are biologically meaningful as there are no statistically analyzed data or compelling text interpretation.
Response: Thanks for your kindly comments, here we only employed the biomolecular fluorescence datasets only for intuitive display of protein nuclear localization and metastasis, these result we used only for qualitative analysis, Not for quantitative analysis.
3. The writing is clear, making the research, results, and conclusions comprehensible. Albeit, the paper should be more hypothesis-based and substantial discoveries more evident.
Response: Thanks for your kindly comments, and we are all glad to get your affirmation. Some of the suggestions you mentioned are indeed right. We will study them in depth later and look forward to the publication of new results

Reviewer 2 Report
The manuscript “Mitogen-activated protein kinases associated sites of tobacco REPRESSION OF SHOOT GROWTH regulates its localization in plant cells” by Wang et al. identified tobacco mitogen-activated protein kinase 3 (NtMPK3) as a RSG interacted protein kinase. Mutation of predicted MAPK-associated phosphorylation site of RSG (Thr-30, Ser-74 and Thr-135) significantly altered the intracellular localization of NtMPK3-RSG interaction complex. This work is well written and could be accepted with minor revision. Here are the comments and suggestions:
1. The supplemental can be moved to the supporting information.
2. A scheme can be added to illustrate the effects of mutations.
Author Response
We appreciate the reviewer’s enthusiasm on our work. Thanks for the detailed comments for clarification and improvement of our manuscript. We have prepared a revised manuscript to follow the reviewer’s concerns. We have addressed all comments below. Besides, we have polished our writing quality by a native English speaker.
- The supplemental can be moved to the supporting information.
Response: We are thankful to the reviewer for making this good suggestion. We have revised this mistake and move supplemental to the supporting information.
- A scheme can be added to illustrate the effects of mutations.
Response: We are thankful to the reviewer for making this good suggestion. We have added a scheme as Figure 7E.

Reviewer 3 Report
Manuscript ID: ijms-1811081
I have reviewed the manuscript by Wang et al., and I have found interesting results and good contributions. However, there are issues that need to be addressed by the authors as indicated below:
Line 23: I suggest the authors to replace “known as one of the important” TFs with “a TF belonging to a bZIP family in tobacco, known to be involved in plant…”
Line 28: I believe the use of …RSG interacting protein… would be fair.
Line 31: change to …after being treated with the plant defense elicitor…within 5 min…within 30 min.
Line 32: I think, “In addition” instead of “moreover” would better indicate the flow and sequence of the ideas here.
Line 36: what do you mean by impaired in its role as TF? (Remove the “s” from TFs). Specify the observed result (was the TF no longer functional or any reduction, inhibition of its activity? Clarify)
Introduction
Line 40: add the definite article “The” and remove “s” from features, and add an “s” to plant (or if you wish to say …“the plant”).
Line 43: by “provokes” do you mean “stress”? If yes, use a simple wording.
Line 44: replace “plenty” with “several” or various. Remove “stages” (line 45) since you have already mentioned aspects earlier.
Lin 46: the authors could rephrase this statement with a simple one: i.e. Previous studies have reported RSG as a…activator…
Line 50: add protein after 14-3-3 and give a short description in brackets.
Line 51: Rephrase the statement and fin alternative words (i.e. it has been suggested that…have been suggested to…)
Lines 85–87: was the objective of the study only to see the possible interaction of between RSG and MAPKs? To which end and under which conditions? Improve the hypothesis statement and clarify.
Results
Lines 98–117: I do not see the importance of presenting this here. In this section, readers expect to see your results at first and how you describe them. Leave the details of other studies in the discussion section. Remove unnecessary statements and focus on describing the data.
Line 114: Do you mean Tobacco nucleus?
Line 115–117: Improve grammar
Line 127: In the figure caption, change to “overexpressing”. Also, indicate the magnification used to visualize the cellular localization of the RSG-CFP and the filters used.
Line 180: correct: it was shown
Line 187: Figure S1 (should move to supplementary material). In addition, the authors should first name the panels (each plate as A, B, and C). Then, rename the subdivisions as 1–6. Afterword, describe properly the panels in the caption. Thereafter, citing the Figure S1 accordingly (Also increase the size of the plates for a clear visibility)
Lines 211–213: I suggest the authors to move these lines after the line 227. The authors can start with the statement in line 214 onward.
Line 330: write transcription factor. Line 331 (A,B) should be PCR and qPCR results. Consider correcting.
Line 336: indicate the number of seeds were sown in each plate
Line 338: write “overexpressing”
Line 341: I recommend the authors to calculate and express this estimate (Figure 7B) as fold change to see the shift in the expression in different backgrounds.
Line 342: Figure 7C does not show the expression of NtGA20ox1 but GA content. Correct this (Figure 7B). In addition, add…the verb between “this might…and…due to…”
Line 345: use “results suggest” instead of “confirmed”
Line 346: write “the germination rate”, and consider rephrasing this sentence.
Discussion
Line 350: I suggest that the authors add a subtitle that deliver a clear message of their findings.
Line 335: write calcium (Ca2+)
Conclusion
The conclusion can be improved to deliver a take-home message from the findings, rather than reproducing the results.
I also think that the conclusion should be placed after the materials and methods section.
Materials and Methods
Line 447: write Agrobacterium tumefaciens and Escherichia coli. Which strains did you used?
Line 448: indicate the growth duration of the culture. Write B. cereus in full name.
Line 450: indicate the growth duration and were? Incubator? Specify if LB broth
Lines 457–458:briefly describe the cloning procedure. Table S2 does not provide a clear understanding on how the authors proceeded or indicate a references or manufacturer’s instructions accordingly.
Lines 503–542: Provide references for each method used (for methods described previously) or give a detail description of the method.
Supplementary material
For a clear visibility of images, the authors should consider increasing the size of Figures S1 and 2.

Author Response
We appreciate the reviewer’s enthusiasm on our work. Thanks for the detailed comments for clarification and improvement of our manuscript. We have prepared a revised manuscript to follow the reviewer’s concerns. We have addressed all comments below. Besides, we have polished our writing quality by a native English speaker.
Line 23: I suggest the authors to replace “known as one of the important” TFs with “a TF belonging toa bZIP family in tobacco, known to be involved in plant…”
Response: We are thankful to the reviewer for making this good suggestion. We have revised “known as one of the important” to “a TF belonging to a bZIP family in tobacco, known to be involved in plant” in the revised manuscript (Line 26-27).
Line 28: I believe the use of …RSG interacting protein… would be fair.
Response: We are thankful to the reviewer for making this good suggestion. We have revised this in the revised manuscript (Line 30).
Line 31: change to …after being treated with the plant defense elicitor…within 5 min…within 30 min.
Response: We are thankful to the reviewer for making this good suggestion. We have revised “known as one of the important” to “a TF belonging to a bZIP family in tobacco, known to be involved in plant” in the revised manuscript (Line 34-35).
Line 32: I think, “In addition” instead of “moreover” would better indicate the flow and sequence of
the ideas here.
Response: We are thankful to the reviewer for making this good suggestion. We have revised “Moreover” to “In addition” in the revised manuscript (Line 35).
Line 36: what do you mean by impaired in its role as TF? (Remove the “s” from TFs). Specify the observed result (was the TF no longer functional or any reduction, inhibition of its activity? Clarify)
Response: We are thankful to the reviewer for making this good suggestion. We have revised this in the revised manuscript (Line 38-39).
Introduction
Line 40: add the definite article “The” and remove “s” from features, and add an “s” to plant (or if you wish to say …“the plant”).
Response: We are thankful to the reviewer for making this good suggestion. We have revised this in the revised manuscript (Line 48).
Line 43: by “provokes” do you mean “stress”? If yes, use a simple wording.
Response: We are thankful to the reviewer for making this good suggestion. We have revised “provokes” to “stress” in the revised manuscript (Line 51).
Line 44: replace “plenty” with “several” or various. Remove “stages” (line 45) since you have already mentioned aspects earlier.
Response: We are thankful to the reviewer for making this good suggestion. We have revised “plenty” to “several”, and removed “stages” in the revised manuscript (Line 52).
Lin 46: the authors could rephrase this statement with a simple one: i.e. Previous studies have reported RSG as a…activator…
Response: We are thankful to the reviewer for making this good suggestion. We have revised this part in the revised manuscript (Line 53-56).
Line 50: add protein after 14-3-3 and give a short description in brackets.
Response: We are thankful to the reviewer for making this good suggestion. We have revised this part in the revised manuscript (Line 56-58).
Line 51: Rephrase the statement and fin alternative words (i.e. it has been suggested that…have been
suggested to…)
Response: We are thankful to the reviewer for making this good suggestion. We have revised this part in the revised manuscript (Line 58-60).
Lines 85–87: was the objective of the study only to see the possible interaction of between RSG and MAPKs? To which end and under which conditions? Improve the hypothesis statement and clarify.
Response: We are thankful to the reviewer for making this good suggestion. We have revised this part in the revised manuscript (Line 90-93).
Results
Lines 98–117: I do not see the importance of presenting this here. In this section, readers expect to see
your results at first and how you describe them. Leave the details of other studies in the discussion
section. Remove unnecessary statements and focus on describing the data. Line 114: Do you mean Tobacco nucleus?
Response: We are thankful to the reviewer for making this good suggestion. We reorganized the Results part and removed unnecessary describes mentioned by Reviewer#3. We have also revised “nuclear” to “nucleus” in the revised manuscript (Line 108).
Line 115–117: Improve grammar
Response: We are thankful to the reviewer for making this good suggestion. We have improved grammar and rewritten mentioned sentence in the revised manuscript (Line 109-111).
Line 127: In the figure caption, change to “overexpressing”. Also, indicate the magnification used to
visualize the cellular localization of the RSG-CFP and the filters used.
Response: We are thankful to the reviewer for making this good suggestion. We have revised this part in the revised manuscript (Line 120-123).
Line 180: correct: it was shown
Response: We are thankful to the reviewer for making this good suggestion. We have revised this part in the revised manuscript (Line 168).
Line 187: Figure S1 (should move to supplementary material). In addition, the authors should first name the panels (each plate as A, B, and C). Then, rename the subdivisions as 1–6. Afterword, describe properly the panels in the caption. Thereafter, citing the Figure S1 accordingly (Also increase the size of the plates for a clear visibility)
Response: We are thankful to the reviewer for making this good suggestion. We have reorganized Figure S1 in the supporting information as required by Reviewer#3.
Lines 211–213: I suggest the authors to move these lines after the line 227. The authors can start with
the statement in line 214 onward.
Response: We are thankful to the reviewer for making this good suggestion. We have revised this part in the revised manuscript (Line 208-210).
Line 330: write transcription factor. Line 331 (A,B) should be PCR and qPCR results. Consider correcting.
Response: We are thankful to the reviewer for making this good suggestion. We have revised this part in the revised manuscript (Line 217-223).
Line 336: indicate the number of seeds were sown in each plate
Response: We are thankful to the reviewer for making this good suggestion. We have added this required descriptions of number of observed seeds in the revised manuscript (Line 235-240)
Line 338: write “overexpressing”
Response: We are thankful to the reviewer for making this good suggestion. We have revised to “overexpressing” in the revised manuscript (Line 228-229).
Line 341: I recommend the authors to calculate and express this estimate (Figure 7B) as fold change to see the shift in the expression in different backgrounds.
Response: We are thankful to the reviewer for making this good suggestion. We have calculated and mentioned the fold change of results in Figure 7B in the revised manuscript (Line 223-225)
Line 342: Figure 7C does not show the expression of NtGA20ox1 but GA content. Correct this (Figure 7B). In addition, add…the verb between “this might…and…due to…”
Response: We are thankful to the reviewer for making this good suggestion. We have corrected this mistake and reorganized mentioned sentence in revised manuscript (Line 231-235).
Line 345: use “results suggest” instead of “confirmed”
Response: We are thankful to the reviewer for making this good suggestion. We have revised “confirmed” to “results suggest” in the revised manuscript (Line 217-223).
Line 346: write “the germination rate”, and consider rephrasing this sentence.
Response: We are thankful to the reviewer for making this good suggestion. We have revised “germinating rate” to “the germination rate” and reorganized this sentence in the revised manuscript (Line 235-238).
Discussion
Line 350: I suggest that the authors add a subtitle that deliver a clear message of their findings.
Response: We are thankful to the reviewer for making this good suggestion. We have added subtitle in the Discussion section to better deliver our messages as required by Reviewer#3.
Line 335: write calcium (Ca2+)
Response: We are thankful to the reviewer for making this good suggestion. We have revised to “calcium (Ca2+)” in the revised manuscript (Line 247).
Conclusion
The conclusion can be improved to deliver a take-home message from the findings, rather than reproducing the results.
Response: We are thankful to the reviewer for making this good suggestion. We have rewritten the Conclusion part as required in the revised manuscript (Line 412-420).
I also think that the conclusion should be placed after the materials and methods section.
Response: We are thankful to the reviewer for making this good suggestion. We have moved the Conclusion part in the revised manuscript (Line 412-420).
Materials and Methods
Line 447: write Agrobacterium tumefaciens and Escherichia coli. Which strains did you used?
Response: We are thankful to the reviewer for making this good suggestion. We have revised to “Agrobacterium tumefaciens” and “Escherichia coli” and added the names of strains in the revised manuscript (Line 332).
Line 448: indicate the growth duration of the culture. Write B. cereus in full name.
Response: We are thankful to the reviewer for making this good suggestion. We have indicated the growth duration of the culture and revised “B. cereus” to “Bacillus cereus” in the revised manuscript (Line 334)
Line 450: indicate the growth duration and were? Incubator? Specify if LB broth
Response: We are thankful to the reviewer for making this good suggestion. We have revised this part in revised manuscript as required (Line 336)
Lines 457–458: briefly describe the cloning procedure. Table S2 does not provide a clear understanding on how the authors proceeded or indicate a references or manufacturer’s instructions accordingly.
Response: We are thankful to the reviewer for making this good suggestion. We have rewritten the cloning procedure in this part and added related reference as required in revised manuscript (Line 343-359).
Lines 503–542: Provide references for each method used (for methods described previously) or give a detail description of the method.
Response: We are thankful to the reviewer for making this good suggestion. We have added related reference as required in revised manuscript (Line 372, 392).
Supplementary material
For a clear visibility of images, the authors should consider increasing the size of Figures S1 and 2.
Response: We are thankful to the reviewer for making this good suggestion. We have improved the visibility of Figures S1 and S2 as required.

Round 2
Author Response
Response to Reviewer 1 Comments
The research focuses on RSG (Regression of Shoot Growth), a leucine zipper transcription factor involved in plant gibberellin feedback regulation. NtMPK3 (CDPK1) phosphorylates RSG at residues Thr-30, Ser-74, and Thr-135, some of which are necessary for RSG nuclear transport. Notably, nuclear transport of RSG, T30A, and S74A occurred with treatment by the flg22 plant defense elicitor, but the mutant versions returned to the cytoplasm within 30 min. These results infer that NtMPK3 is necessary for nuclear localization of RSG by flg22 for plant defensive function.
The experimentation is comprehensive and highly technical, and the results thoroughly and adequately support the stated conclusions in the manuscript. Criticism is that the research is not very hypothesis-based, and the investigation is an enormous biochemical and genetic tactical endeavor to decipher RSG function with associated partners molecules. Principle hypotheses about RSG’s critical biological functions in plant defense derive mainly from experimentation that utilizes heterologous biological systems for functional tests but can not directly establish cause and effect biology between the plant and the microbe. The yeast genetic experiments seem adequately conducted but not directly interpretable. The biomolecular fluorescence datasets do not confirm whether the sample differences are biologically meaningful as there are no statistically analyzed data or compelling text interpretation.
The writing is clear, making the research, results, and conclusions comprehensible. Albeit, the paper should be more hypothesis-based and substantial discoveries more evident.
We appreciate the reviewer’s enthusiasm on our work. Thanks for the detailed comments for clarification and improvement of our manuscript. We have prepared a revised manuscript to follow the reviewer’s concerns. We have addressed all comments below.
- The experimentation is comprehensive and highly technical, and the results thoroughly and adequately support the stated conclusions in the manuscript. Criticism is that the research is not very hypothesis-based, and the investigation is an enormous biochemical and genetic tactical endeavor to decipher RSG function with associated partners molecules. Principle hypotheses about RSG’s critical biological functions in plant defense derive mainly from experimentation that utilizes heterologous biological systems for functional tests but can not directly establish cause and effect biology between the plant and the microbe.
Response: First of all, thank you for your affirmation of this study. In this study, we identified tobacco mitogen-activated protein kinase 3 (NtMPK3) as a RSG interacting protein kinase. And demonstrated that MAPK associated phosphorylation sites of RSG regulate its localization in tobacco and constant unphosphorylation of RSG in Thr-30, Ser-74 and Thr-135 keeps RSG predominantly localized in cytoplasm. Indeed, the above findings and conclusions are mainly proved by biochemical research methods, lack of further biological verification, and we simplely discuss the impact of RSG by some immunity elicitor such as flg22, PGPR strain et al. Laterly, we will integrate more biological methods to study the function of related genes in the interaction between plants and pathogens.
- The biomolecular fluorescence datasets do not confirm whether the sample differences are biologically meaningful as there are no statistically analyzed data or compelling text interpretation.
Response: Thanks for your kindly comments, here we only employed the biomolecular fluorescence datasets only for intuitive display of protein nuclear localization and metastasis, these result we used only for qualitative analysis, Not for quantitative analysis.
- The writing is clear, making the research, results, and conclusions comprehensible. Albeit, the paper should be more hypothesis-based and substantial discoveries more evident.
Response: Thanks for your kindly comments, and we are all glad to get your affirmation. Some of the suggestions you mentioned are indeed right. We will study them in depth later and look forward to the publication of new results
Reviewer 3 Report
The authors have addressed almost all the concerns raised in the round of revision. The manuscript has been significantly improved. The authors should pay attention to some wordings that may mislead the readers. In lines 463 and 466, the authors could use...would impair seed germination... instead of bringing damage.
In addition, the authors should consider replacing ...take an example by bZIP family... with a more meaningful wording to clearly deliver their message.
Author Response
The authors have addressed almost all the concerns raised in the round of revision. The manuscript has been significantly improved. The authors should pay attention to some wordings that may mislead the readers. In lines 463 and 466, the authors could use...would impair seed germination... instead of bringing damage.
In addition, the authors should consider replacing ...take an example by bZIP family... with a more meaningful wording to clearly deliver their message.
Response: We are thankful to the reviewer for making this good suggestion. we have revised the manuscript according to your comments. and recheck the paper carefully and modified all part wich may mislead the readers. we have use"would impair seed germination"instead of" bringing damage". Moreover, we have replaced "take an example by bZIP family" and rewriting the sentence.